# Biophysical Characterization of the Binding Mechanism between the MATH Domain of SPOP and Its Physiological Partners

**DOI:** 10.3390/ijms241210138

**Published:** 2023-06-14

**Authors:** Awa Diop, Paola Pietrangeli, Caterina Nardella, Valeria Pennacchietti, Livia Pagano, Angelo Toto, Mariana Di Felice, Sara Di Matteo, Lucia Marcocci, Francesca Malagrinò, Stefano Gianni

**Affiliations:** Laboratory Affiliated to Istituto Pasteur Italia-Fondazione Cenci Bolognetti, Dipartimento di Scienze Biochimiche “A. Rossi Fanelli”, Sapienza Università di Roma, P.le Aldo Moro 5, 00185 Rome, Italy; awa.diop@uniroma1.it (A.D.); paola.pietrangeli@uniroma1.it (P.P.); caterina.nardella@uniroma1.it (C.N.); valeria.pennacchietti@uniroma1.it (V.P.); livia.pagano@uniroma1.it (L.P.); angelo.toto@uniroma1.it (A.T.); mariana.difelice@uniroma1.it (M.D.F.); sara.dimatteo@outlook.it (S.D.M.); lucia.marcocci@uniroma1.it (L.M.); francesca.malagrino@unina.it (F.M.)

**Keywords:** kinetics, protein–protein interactions, MATH domain, speckle-type POZ protein

## Abstract

SPOP (Speckle-type POZ protein) is an E3 ubiquitin ligase adaptor protein that mediates the ubiquitination of several substrates. Furthermore, SPOP is responsible for the regulation of both degradable and nondegradable polyubiquitination of a number of substrates with diverse biological functions. The recognition of SPOP and its physiological partners is mediated by two protein–protein interaction domains. Among them, the MATH domain recognizes different substrates, and it is critical for orchestrating diverse cellular pathways, being mutated in several human diseases. Despite its importance, the mechanism by which the MATH domain recognizes its physiological partners has escaped a detailed experimental characterization. In this work, we present a characterization of the binding mechanism of the MATH domain of SPOP with three peptides mimicking the phosphatase Puc, the chromatin component MacroH2A, and the dual-specificity phosphatase PTEN. Furthermore, by taking advantage of site-directed mutagenesis, we address the role of some key residues of MATH in the binding process. Our findings are briefly discussed in the context of previously existing data on the MATH domain.

## 1. Introduction

The regulation of protein homeostasis is a critical biological process that relies on protein translation, chaperone-assisted folding, and protein degradation pathways [1,2,3]. In this context, the ubiquitination-mediated degradation pathway plays a pivotal role, constituting the major protein degradation pathway in the cell, with the ubiquitin proteasome system controlling approximately 80% of the degradation of intracellular proteins [4,5]. A critical step in ubiquitin-mediated degradation is exerted by E3 ligases, which actively tag proteins with ubiquitin to be subsequently recognized by the proteasome for digestion and fragmentation. Because of their importance and specific target recognition, E3 ligases are very interesting therapeutic targets, being deregulated in several human diseases [6,7,8]. 

SPOP (Speckle-type POZ protein) is an E3 ubiquitin ligase adaptor protein that mediates the ubiquitination of several substrates and was originally discovered in 1997 by Nagai and coworkers [9]. SPOP actively orchestrates several biological processes by specifically recognizing different substrates that are involved in diverse mechanisms, ranging from chromatin organization, as in the case of the chromatin organizer and epigenomic regulator MacroH2A [10], and cell metabolism, in the case of Phosphatase and Tension homologue (PTEN) protein [11]. In fact, PTEN (Phosphatase and Tensin homologue) is involved in the regulation of the cell cycle, because of its tumor suppressor activity and its role as a negative regulator of the AKT/NF-KB signaling pathway [12]. Of additional relevance, SPOP can also bind and regulate the protein Puc, a Jun-kinase phosphatase puckered that is highly expressed in most clear cell renal cell carcinoma [13].

From a structural perspective, SPOP is characterized by a modular nature, comprising an N-terminal portion, containing a Meprin and TRAF-C Homology (MATH) domain, followed by a BTB domain, a 3-box, and an NLS disordered motif at its C-terminal part. Importantly, whilst the BTB domain facilitates the formation of the complex with CUL3, leading to the active form of the enzyme [14], the MATH (Meprin and TRAF-C Homology) domain mediates the recognition of the target substrates to be ubiquitinated and, therefore, it is of critical importance for the correct functions of the SPOP-mediated degradation pathways. Notably, mutations on the MATH domain of the SPOP protein have been reported as responsible for certain types of cancers such as prostate and endometrial cancers. In the case of breast and brain cancers, a loss of protein expression is observed, whereas for kidney and ovary cancers, an overexpression is noted [15].

MATH domains typically consist of seven β-strands forming an antiparallel β-sandwich architecture [13]. The three-dimensional structure of the MATH domain of SPOP is reported in Figure 1 (pdb code 2CR2). As briefly recalled above, the MATH domain of SPOP is a multifunctional binding domain which can selectively recognize different physiological partners. The MATH domain interacts with SPOP substrates via a conserved consensus motif in the substrates named an SBC (SPOP-binding consensus) motif. The SBC motif consists of a nonpolar residue followed by a polar residue, a serine, and then a threonine (ϕ-π-S-S/T-S/T) [13]. 

Previous structural work suggested that the interaction between the MATH domain and the SBC motif is primarily mediated by a cluster of residues, constituting its binding pocket [13]. However, despite these indications, to date, no biophysical work has investigated the underlying mechanism of the recognition of MATH domains. Hence, with the aim of deciphering the binding mechanism of SPOP MATH to its partners, we resorted here to providing a complete characterization of the binding properties of this domain both through equilibrium and time-resolved, stopped-flow kinetic experiments. We used the engineered wildtype form, called SPOP MATH WT, where the four cysteine residues are substituted to serine, in order to decipher the binding mechanism of the monomeric domain. As detailed below, to address binding, we employed peptides mimicking the physiological binders Puc, MacroH2A, and PTEN. Furthermore, by taking advantage of site-directed mutagenesis, we critically investigated the role of the residues located in the binding pocket R71, Y88, Y124, K130, D131, W132, F134, and K135 in defining the binding affinity of SPOP MATH to its physiological partners. Our findings are discussed in light of previous works carried out on the SPOP MATH domain. 

## 2. Results

### 2.1. Characterization of SPOP MATH Binding Mechanism

The MATH domain of SPOP exerts its functions by specifically recognizing different substrates and, therefore, acting as an adaptor module between SPOP and its physiological ligands [16]. In an effort to provide a complete characterization of SPOP MATH binding mechanisms, we investigated the recognition of three peptides mimicking a specific region of the substrates Puc (region 96–105), MacroH2A (region 166–179), and PTEN (region 354–368). Notably, all three peptides contain the SBC motifs, while displaying a different amino acidic sequence. Moreover, the three-dimensional structure of SPOP MATH in complexes with each of the peptides has been previously reported (3HQH, 3HQL, and 4OV1 PDB entries, respectively). Similar to previous work on other protein–protein recognition modules [17,18,19], we employed peptides carrying a covalently attached N-terminal dansyl fluorophore. Binding was then followed by measuring the FRET (Forster resonance energy transfer) between the three tryptophan residues of the SPOP MATH domain (donor) and the dansyl group of the peptides (acceptor). 

To address the equilibrium binding affinity of SPOP MATH and its substrates, a fixed concentration of the protein (1 µM) was challenged with increasing peptide concentrations. In all cases, the fluorescence-monitored transition returned a hyperbolic behavior as described in the Section 4, which allowed estimating a dissociation constant of 1.4 ± 0.4 µM, 1.6 ± 0.2 µM, and 5.5 ± 0.5 µM for Puc, MacroH2A, and PTEN, respectively (Figure 2). These values appear partly consistent with what was previously reported by Zhuang et al. [13] by using surface plasmon resonance. We note that the minor disagreement observed between our measurements and those previously reported by Zhuang et al. may have arisen from the immobilization to a solid surface, which is applied in surface plasmon resonance experiments, which may have partly jeopardized the analysis when working with relatively small protein domains and/or peptides.

To infer the mechanism of recognition of SPOP MATH, we resorted to performing time-resolved kinetic experiments. SPOP MATH was rapidly mixed with increasing concentrations of dansylated peptides (from 2 to 10 µM for Puc and MacroH2A, and from 2 to 40 µM for PTEN) in a stopped-flow apparatus at 298 K. The binding traces obtained by monitoring the time-resolved fluorescence were satisfactorily fitted to a single exponential equation, allowing calculating the observed rate constants *k_obs_* at different peptide concentrations. We then plotted the *k_obs_* values over the concentration of dansylated peptide and fitted the data to a simple linear function equation:$$k_{obs}=k_{on}\;\left[Peptide\right]+k_{off}$$
where *k_obs_* is the observed rate constant; *k_on_* and *k_off_* are the microscopic association and dissociation rate constants, respectively (Figure 2). The slope and y-axis intercept represent the microscopic association and dissociation rate constant, respectively. To minimize the error associated with the indirect measurement of the dissociation rate constant, we carried out displacement experiments to determine the *k_off_* values. The displacement experiments were designed by mixing the preformed complex SPOP MATH-dansylated peptides with a high excess of nondansylated peptide, as described in the Section 4. By combining the association rate constants calculated from the analysis of the pseudo-first-order plots with the dissociation rate constants obtained from the displacement experiments, the calculated K_D_ for Puc, MacroH2A, and PTEN was 1.4 ± 0.4 µM, 7.8 ± 0.4 µM, and 20.8 ± 0.4 µM, respectively. Notably, whilst the value of K_D_ calculated for Puc was in very good agreement with that obtained from the equilibrium experiments, in the case of both MacroH2A and PTEN, the affinity observed for equilibrium was higher than that estimated from kinetics (Table 1). This finding suggests that, whilst the binding of Puc is consistent with a two-state scenario, in the case of MacroH2A and PTEN, there is a deviation from such mechanisms. On the basis of these observations, it is possible to conclude that, in the case of binding to MacroH2A and PTEN, there is the presence of an additional step, which escapes detection by stopped-flow experiments and is likely to be negligible in the case of Puc. Notably, while the disagreement between equilibrium and kinetic experiments observed for MacroH2A and PTEN allows excluding two-state binding mechanisms, since we could not directly observe any additional kinetic phases and the observed rate constants varied linearly with the varying peptide concentration, it is not possible at this stage to infer the mechanism further. We note that such a case closely resembles the presence of a so-called burst phase, which is frequently observed in protein-folding studies.

### 2.2. Increasing Ionic Strength Weakens the Affinity of SPOP MATH to Its Substrates

Previous structural characterization of the interaction between SPOP MATH and its partners suggested the binding site involved several charged groups [13]. Hence, to elucidate the effect of electrostatic interactions in the complex formation, we performed binding experiments at different ionic strengths. The equilibrium and kinetic behavior of SPOP MATH binding recorded in the presence of increasing concentrations of NaCl is reported in Figure 3. In agreement with previous evidence, we observed that increasing ionic strength resulted in a detectable decrease in binding affinity for all peptides. Remarkably, in the case of PTEN, such behavior was very pronounced, such that we could not measure any binding when the concentration of NaCl was higher than 50 mM. Conversely, in the case of Puc and MacroH2A, binding experiments could be performed up to 500 mM NaCl. The calculated binding affinities, as well as the association and dissociation rate constants for SPOP MATH and its physiological binders as obtained from the different experiments reported in Figure 3, are listed in Appendix A. 

Inspection of the data reported in Figure 3 and Appendix A indicates that, for SPOP MATH, the decrease in affinity with increasing ionic strength was primarily due to a decrease in the association microscopic rate constant *k*_on_, whereas, on the whole, *k*_off_ remained unchanged. On the basis of these observations, it may be concluded that electrostatic interactions play an important role in the formation of the complex by stabilizing the interactions in the transition state, with a very pronounced effect in the case of Puc. Interestingly, the net charge of both Puc and PTEN peptides is identical, both displaying two acidic groups, while showing a clearly different ionic strength dependence. This finding suggests that the distribution of charges around the SBC motif might be critical in dictating the electrostatic interactions stabilizing the transition state of binding. 

Finally, it is of interest to observe that—analogous to what was noted above for the data recorded at a low ionic strength—whilst in the case of Puc, the data are consistent with a two-state binding mechanism with a very good agreement between the K_D_ calculated from equilibrium and kinetic experiments, in the case of both MacroH2A and PTEN, there is a detectable difference. Such observations reinforce our indication that the binding of these two peptides occurs via a more complex scenario. 

### 2.3. Analyzing the Binding of SPOP MATH by Site-Directed Mutagenesis

To quantitatively address the role of specific residues in the binding mechanism of SPOP MATH, we resorted to taking advantage of the site-directed mutagenesis and generated ten site-directed mutants corresponding to the residues located in the binding pocket of the SPOP MATH domain. In particular, we produced a set of single mutants where residues R71, Y88, F103, Y124, F125, K130, D131, W132, F134, and K135 were substituted with alanine. Importantly, those residues were previously suggested to interact directly with the physiological binders of SPOP MATH, through the conserved consensus motif that consists of φ-π-S-S/T-S/T. 

The dependence of the binding rate constant of the different mutant variants compared to the results obtained for the wildtype SPOP MATH is reported in Figure 4. Unfortunately, in many cases, the affinities of the mutants were too low to allow an experimental detection and we could not observe any binding. This finding confirms the earlier observations suggesting that, by and large, these positions are critical for the recognition of the specific partners of SPOP MATH. Table 2 summarizes the obtained results.

Because of the low affinity, it was very difficult to quantitatively analyze the mutational work for the peptides mimicking PTEN and MacroH2A. On the other hand, in the case of Puc, it is of interest to analyze the structural distribution of the observed results. Figure 5 highlights the effects of the mutational work, in the case of the Puc peptide, mapped onto the structure of SPOP MATH. It is evident that positions Y88, K130, and D131 are clustered within the binding pocket surrounding the C-terminal part of the consensus motif. These mutations, reported in red in the figure, primarily affect the dissociation rate constant, indicating that the stabilizing effect of these positions takes place downhill from the main binding transition state. Conversely, positions Y124, F134, and K135 destabilize binding by affecting both the association and dissociation rate constants, thereby indicating these residues have a direct role in both transition state and ground state stabilization. Interestingly, these positions appear more sparsely distributed around the binding pocket, possibly suggesting that the initial encounter between SPOP MATH and its partners occurs via an extended conformation that subsequently locks in the complex ground state. 

## 3. Discussion

Understanding the mechanisms by which protein–protein interaction modules interact with their partners is of critical importance to define their functions as well as to pave the way for the design of specific inhibitors. From this perspective, despite their critical importance and despite being present in hundreds of different eukaryotic proteins, MATH domains have, to date, eluded detailed experimental characterization. In this work, by taking advantage of a versatile MATH domain, recognizing at least three distinct physiological ligands, we resorted to analyzing in detail the equilibrium and kinetic properties of binding of a prototypical example of MATH domains, SPOP MATH.

A common method to describe the properties of the binding mechanism between interacting molecules lies in comparing the thermodynamic parameters obtained from different experimental approaches [20]. In the case of SPOP MATH, it appears that, whilst in the case of the interaction with Puc, the K_D_ extracted from equilibrium and kinetic experiments returned comparable values, for MacroH2A and PTEN, there were detectable differences. In fact, in these cases, the affinity of the complexes appeared higher when calculated from equilibrium experiments as compared to that observed from kinetics. On the basis of these observations, it is possible to conclude that, whilst the binding between SPOP MATH and Puc occurs in an all-or-none two-state fashion, in the case of the other partners, there is the presence of at least one intermediate. Notably, however, since we could neither observe any deviation from linearity in the dependence of the observed rate constant nor detect any additional kinetic phase, it is important to stress that, at this stage, the binding mechanism between MATH and Puc cannot be described in further detail.

A binding mechanism processing through the accumulation of an intermediate is often linked to detectable conformational changes in the interacting partners [21,22,23,24,25]. From this perspective, it is of interest to compare the structure of SPOP MATH in complex with Puc, MacroH2A, and PTEN, which have been previously resolved [13] (Figure 6). Interestingly, it appears that SPOP MATH is very robust, and in all three complexes, its structure appears essentially identical and readily superposable with the structure of MATH determined in isolation (Figure 6). This finding would suggest that the slide in binding mechanism, from two-state to multistate, is unlikely to arise from a conformational change in SPOP MATH and may, therefore, be ascribed to a multistep recognition event of the different peptides. From this perspective, however, it is important to stress that whilst a detectable change in conformation could not be highlighted from the structural superposition reported in Figure 6, it is not possible at this stage to exclude changes in the dynamic properties of the domain upon binding. 

It is of interest to discuss the mutational data of SPOP MATH in light of the experiments carried out at different ionic strengths. In fact, for all three peptides, we observed a pronounced decrease in the calculated association rate constants with increasing ionic strength. This observation indicates that, despite the fact that the SPOP-binding motif does not directly imply the presence of charged residues (being φ-π-S-S/T-S/T; where φ is a nonpolar residue, and π is a polar residue), the transition state is primarily stabilized by the electrostatic interactions between SPOP MATH and its binding partners, indicating that the charges, as well as their physical distribution around the SPOP-binding motif, are critical for binding. From this perspective, it is of interest to note that positions Y88, K130, and D131, which are clustered in the center of the binding pocket, primarily affect the dissociation rate constants. On the other hand, positions Y124, F134, and K135 destabilize binding by affecting both the association and dissociation rate constants, thereby indicating these residues have a direct role in both transition state and ground state stabilization. Overall, our data support a mechanism whereby the early stages of binding are primarily dictated by electrostatic interactions between SPOP MATH and its binding partners, which is followed by a subsequent locking of the specific interactions located in the binding site of the domain. Future work with other members of the MATH domain family will clarify the general nature of these observations. 

## 4. Materials and Methods

### 4.1. Protein Expression and Purification

A construct encoding the engineered MATH domain of SPOP protein (residues 1–182), with cysteine residues substituted to serine, was subcloned in a pHTP1 plasmid vector and then transformed in *Escherichia coli* cells BL21 (DE3). Bacterial cells were grown in LB medium, containing 30 μg/mL of kanamycin, at 37 °C until OD_600_ = 0.7–0.8, and then protein expression was induced with 1mM IPTG. After induction, cells were grown at 25 °C overnight and then collected using centrifugation. To purify the His-tagged protein, the pellet was resuspended in buffer made of 50 mM Tris-HCl, 300 mM NaCl, and 10 mM Imidazole, with a pH of 7.5 and with the addition of antiprotease tablet (cOmplete, EDTA-free, Roche Diagnostics GmbH, Mannheim, Germany), then sonicated and centrifuged. The soluble fraction from bacterial cell lysate was loaded onto a nickel-charged HisTrap Chelating HP (GE Healthcare Bio-Sciences AB, Uppsala, Sweden) column equilibrated with 50 mM Tris-HCl, 300 mM NaCl, 10 mM Imidazole, and a pH of 7.5. The protein was then eluted with a gradient from 0 to 1 M imidazole by using an ÄKTA-prime system. Fractions containing the protein were collected and the buffer was exchanged to 50 mM Tris-HCl and 300 mM NaCl, pH 7.5. The purity of the protein was analyzed through SDS-PAGE. Site-directed mutagenesis was performed using the NZYTech mutagenesis kit (NZYTech, Lisbon, Portugal), according to the manufacturer’s instructions. Peptides mimicking SPOP MATH substrates, Puc (sequence DEVTSTTSSS), MacroH2A (sequence KAASADSTTEGTPAD), and PTEN (sequence PSNPEASSSTSVTPD), with and without the dansyl N-terminal modification, were purchased from GenScript (GenScript Biotech, Rijswijk, Netherlands). 

### 4.2. Equilibrium Binding Experiments

Equilibrium binding experiments were performed on a Fluoromax-4 single-photon-counting spectrofluorometer (Jobin-Yvon, Newark, NJ, USA). The SPOP MATH domain was excited at 280 nm and emission spectra were recorded between 300 and 400 nm, at increasing dansylated peptide concentrations. Experiments were performed with the engineered SPOP MATH wildtype at a constant concentration of 2 µM and titrated with increasing concentrations of dansylated peptides from 0 to 40 µM at 298 K, using a quartz cuvette with a path length of 1 cm. Data were fitted using the following hyperbolic function:$$Y_{obs}=\frac{Y_{max}\cdot \left[peptide\right]}{K_{D}+\left[peptide\right]}+costant$$

### 4.3. Stopped-Flow Binding Experiments

Binding kinetic experiments were performed on an Applied Photophysics DX-17MV stopped-flow apparatus (Applied Photophysics, Leatherhead, UK). Pseudo-first-order binding experiments were performed by mixing a constant concentration (2 μM) of MATH domain with increasing dansylated peptides concentrations, from 2 to 10 μM of Puc and MacroH2A, and 2 to 40 µM of PTEN. The samples were excited at 280 nm, and the emission fluorescence was recorded by using a bandpass 320 nm cutoff filter. The experiments were performed at 298 K. For ionic strength dependence with Puc and MacroH2A, the buffers used were 50 mM sodium-HEPES, pH 7.2, supplemented with 50 mM, 100 mM, 200 mM, or 500 mM of sodium chloride (NaCl). For the PTEN binding salt dependence experiments, lower concentrations of salt were added for the previous buffer list mentioned (10 mM, 25 mM, and 50 mM of sodium chloride). For each acquisition, five traces were collected, averaged, and satisfactorily fitted to a single exponential equation.

### 4.4. Stopped-Flow Displacement Experiments

Microscopic dissociation rate constants (*k_off_*) were directly measured by performing displacement experiments on an Applied Photophysics DX-17MV stopped-flow apparatus (Applied Photophysics). A preincubated complex of MATH domain and dansylated peptides at a constant concentration (both 8 µM) was rapidly mixed with an excess of nondansylated peptides (50 μM). The samples were excited at 280 nm and fluorescence emission was collected by using a 330 ± 30 nm bandpass filter. The experiments were performed at 298 K in the same buffer used for the binding experiments. The observed rate constants were calculated from the average of five single traces. The observed kinetics were consistent with a single exponential decay. A typical binding and displacement transition is reported in Appendix A.

## Figures and Tables

**Figure 1 ijms-24-10138-f001:**
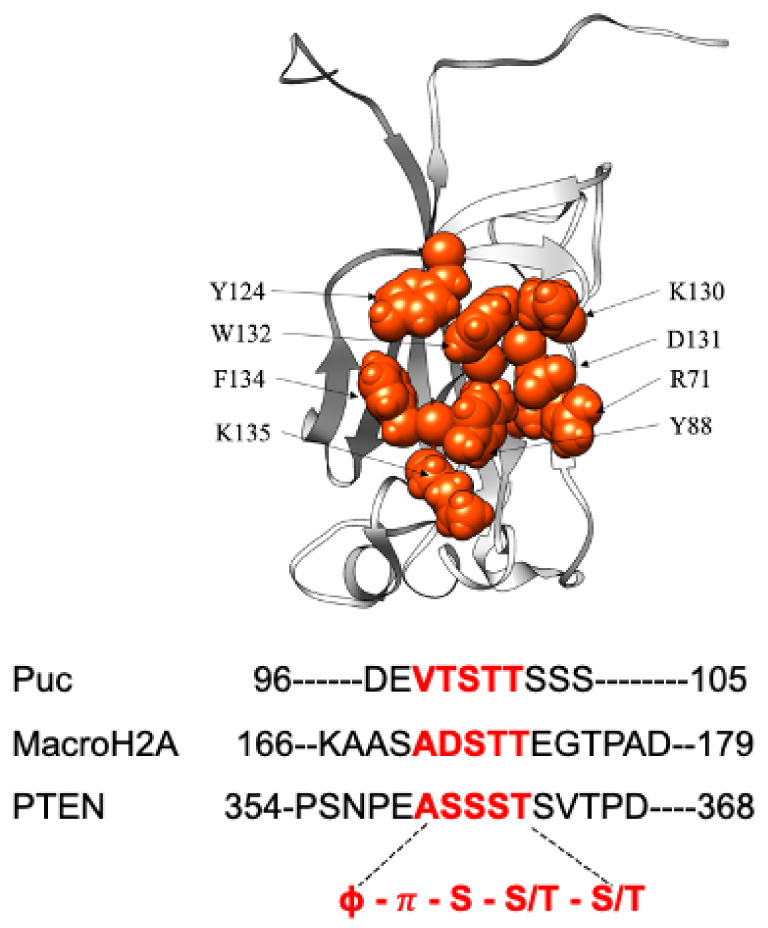
Three-dimensional structure of SPOP MATH domain (PDB: 2CR2). A sequence alignment of the peptides Puc, MacroH2A, and PTEN, highlighting the consensus motif in red, is presented below image.

**Figure 2 ijms-24-10138-f002:**
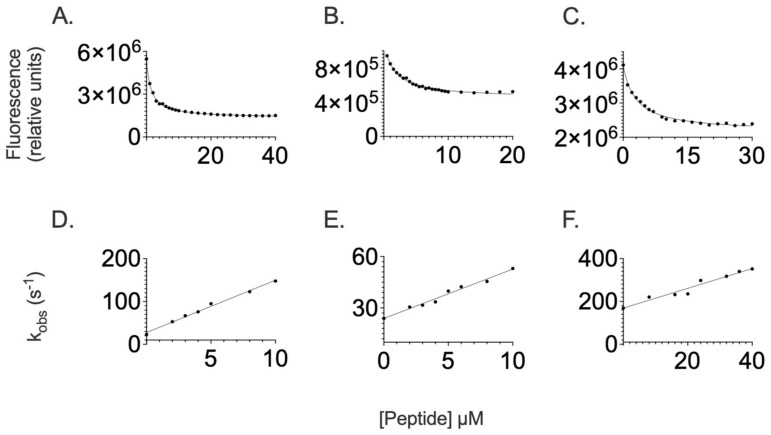
Binding experiments between the SPOP MATH domain and peptides mimicking Puc, MacroH2A, and PTEN. Equilibrium binding experiments between SPOP MATH WT and dansylated peptides Puc (panel (**A**)), MacroH2A (panel (**B**)), and PTEN (panel (**C**)). Curves represent the best fit to a hyperbolic equation. Binding kinetics experiments between SPOP MATH WT and dansylated peptides Puc (panel (**D**)), MacroH2A (panel (**E**)), and PTEN (panel (**F**)). The dependences of *k_obs_* as a function of peptide concentrations were fitted to a linear equation. The related calculated kinetic parameters are listed in Table 1.

**Figure 3 ijms-24-10138-f003:**
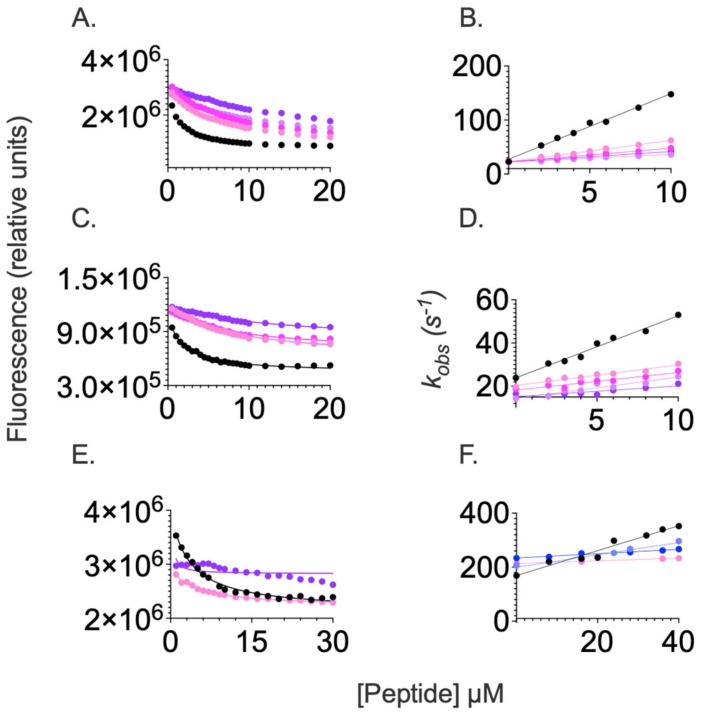
Binding equilibrium and kinetic experiments between SPOP MATH and dansylated peptides Puc (panel (**A**,**B**)), MacroH2A (panel (**C**,**D**)), and PTEN (panel (**E**,**F**)) performed at different concentrations of sodium chloride ranging from 0 mM to 500 mM. Lines represent the best fit to a hyperbolic function (for equilibrium experiments) and straight line (for kinetic binding experiments). 0mM NaCl (black); 15 mM NaCl (violet); 25 mM NaCl (blue); 50 mM NaCl (light magenta); 100 mM NaCl (magenta); 200 mM NaCl (lavender); and 500 mM NaCl (purple).

**Figure 4 ijms-24-10138-f004:**
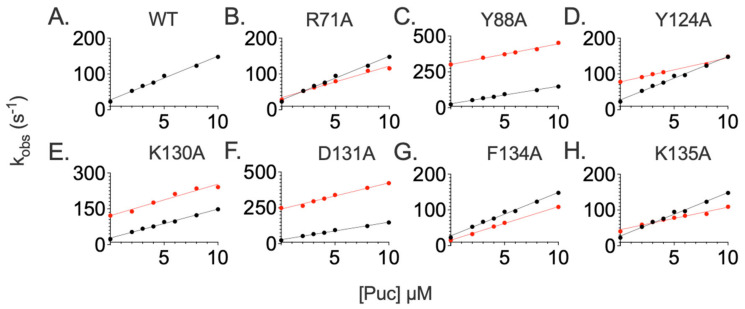
Kinetic binding experiments between SPOP MATH WT (panel (**A**)) in black, and site-directed variants R71A (panel (**B**)), Y88A (panel (**C**)), Y124A (panel (**D**)), K130A (panel (**E**)), D131A (panel (**F**)), F134A (panel (**G**)), and K135A (panel (**H**)). For each panel, the black and red dots correspond to the binding with the WT or mutant variant of SPOP MATH domain. Lines represent the best fit to a linear equation. The related kinetic parameters are listed in Table 2.

**Figure 5 ijms-24-10138-f005:**
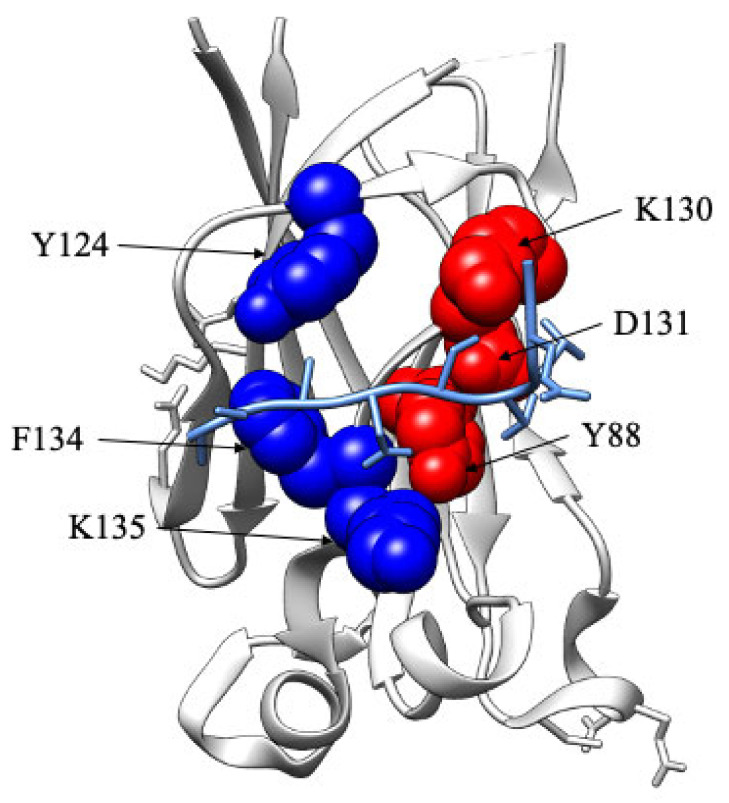
Mapping the binding pocket of SPOP MATH WT, in complex with Puc (pdb code: 3HQH). The amino acid residues affecting the dissociation rate constant (*k_off_*) and both association (*k_on_*) and dissociation (*k_off_*) rate constants are highlighted in red and blue, respectively.

**Figure 6 ijms-24-10138-f006:**
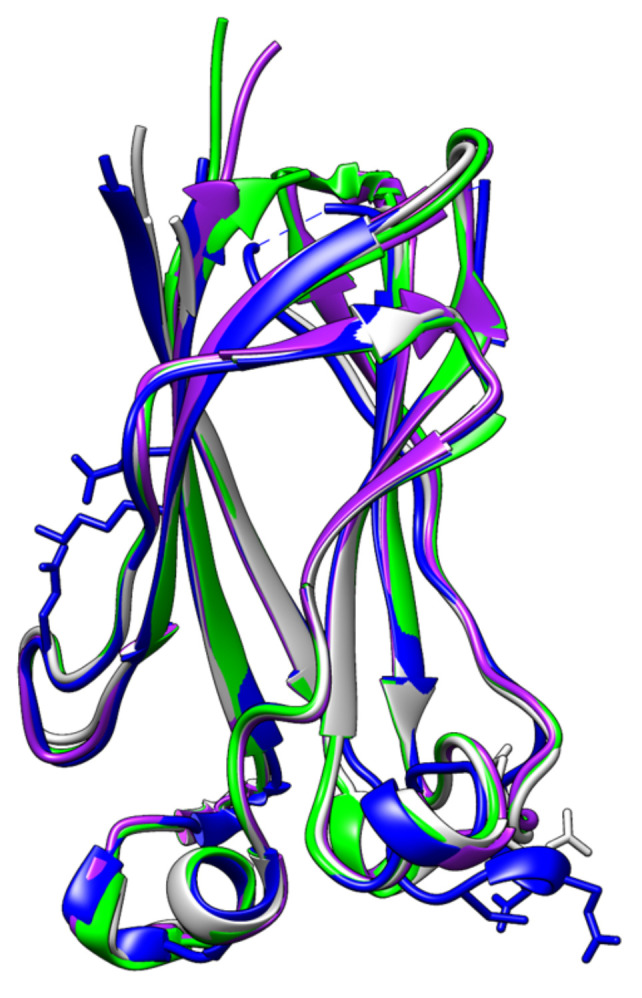
Superimposed structures of SPOP MATH WT in its apo form and in complex with peptides Puc, MacroH2A, and PTEN. PDB entries used were 7KPI (apo SPOP MATH, in purple), 3HQl (complex SPOP MATH:Puc, in light gray), 3HQH (complex SPOP MATH: MacroH2A, in blue), and 4OV1 (complex SPOP MATH: PTEN, in green).

**Table 1 ijms-24-10138-t001:** Equilibrium and binding kinetics parameters of SPOP MATH WT with the dansylated peptides mimicking Puc, MacroH2A, and PTEN.

	Equilibrium	Kinetics		
	*K_D_* (µM)	*k_on_* (µM^−1^ s^−1^)	*k_off_* (s^−1^)	*K_D_* (µM)
Puc	1.4 ± 0.5	19 ± 1	26.1 ± 0.1	1.4 ± 0.4
MacroH2A	1.6 ± 0.2	3 ± 2	21 ± 2	7.8 ± 0.4
PTEN	4 ± 1	5.5 ± 0.5	113 ± 1	20.8 ± 0.4

The calculated parameters were obtained using a hyperbolic and a pseudo-first-order analysis in the case of equilibrium and kinetic data, respectively, as described in the Section 4.

**Table 2 ijms-24-10138-t002:** Binding kinetics parameters for SPOP MATH mutant variants’ interaction with dansylated Puc.

MATH Variants		Puc	
	*k_on_* (µM^−1^ s^−1^)	*k_off_* (s^−1^)	*K_D_* (µM)
WT	19 ± 1	26.1 ± 0.1	1.4 ± 0.4
R71A	8.8 ± 0.5	34 ± 3	3.8 ± 0.1
Y88A	14.4 ± 0.8	302 ± 5	21 ± 4
Y125A	6.71 ± 0.02	78.0 ± 0.2	11.7 ± 0.2
K130A	13 ± 2	123 ± 16	9.5 ± 0.1
D131A	18.2 ± 0.1	241 ± 1	13.3 ± 0.2
W132A	N/A	N/A	N/A
F134A	9.4 ± 0.2	15 ± 1	1.7 ± 0.1
K135A	6.3 ± 0.5	45.2 ± 0.4	7.2 ± 0.4

## Data Availability

The datasets generated and/or analyzed in this study are available from the corresponding author upon reasonable request.

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
