# Peer review of "Biophysical Characterization of the Binding Mechanism between the MATH Domain of SPOP and Its Physiological Partners"

_ijms, 2023, doi:10.3390/ijms241210138_

Round 1
Reviewer 1 Report
The study by Diop et al describes the detailed characterization of the peptide binding of the MATH domain of SPOP. The study provides information on the affinities of three selected ligands, and some insights into binding mechanisms that may be of interest for the readers of IJMS.
There are some minor things to be corrected:
· Abstract
There is something wrong with the sentence ” The complex functions of SPOP are mediated by its modular nature, which comprise different protein-protein interaction domains. ” It is not clear what is intended with “The complex function” and why that should be mediated by its “modular nature, which comprise different protein-protein interaction domains. I would suggest that the authors rephrase, or at least split the sentence into two.
By the way, SPOP has only two folded domains, so the authors may consider being precise with the information.
· Introduction
Check the text for minor mistakes. I list same but not all here:
Introduce the full name and abbreviation of PTEN at first time use, not in the following sentence. Same thing for MATH.
The NLS is not a structured domain, it is a motif in an intrinsically disordered region, I assume.
The alpha sign has become @ in alpha-strand in multiple places.
· Results
Line 85-86.
Something is wrong in the sentence “SPOP MATH exerts its functions by specifically recognizing different substrates and, therefore, acting as an adaptor module between SPOP and its physiological ligands [16].” SPOP is the adaptor, hence the domain cannot act as an adaptor module between the protein itself and its ligands. Please rephrase.
Line 88-89
In the sentence “a specific region of the sub-strates Puc, MacroH2A, and PTEN “ the authors should indicate the amino acid regions for each protein.
Line 98:
Space is missing ”(1μM)”.
Figure 3. The resolution is low, lines are fuzzy. Please ensure high resolution figures.
Discussion:
Please mention how many MATH domains there are in humans to give a bit of context.
Figure 6. Figure legend is odd. “SPOP MATH WT-PEPTIDES” , please modify.
Experiments
What is the origin of the engineered MATH domain? Synthetic? If so indicate from which company.
Reviewer 2 Report
The manuscript by Diop et al. describes binding experiments between the SPOP MATH domain and three physiological peptide ligands, puc, MacroH2A and PTEN. Structural and binding studies of SPOP MATH to puc and MacroH2A have been reported in ref. 13, including mutational analysis identical to that reported in this study. Thus, the novelty of this study is diminished, besides the PTEN work, because the mechanism of binding has already been described by atomic resolution structural data and mutational analysis (ref. 13). Much of the work presented in this manuscript augments or verifies previous work with limited new insight into the binding mechanism of peptide ligands to the SPOP MATH domain. The kinetics analysis is a strength of the work presented, as is the possible observation of a non-two-state binding model (although that requires further experimental evidence before making such a speculative claim). The following comments should be considered in any future submission:
(i) An alignment of the three peptides, puc, MacroH2A and PTEN should be included, perhaps in Figure 1. Figure 1 can be improved by highlighting in color and residue labeling the peptide binding site, e.g., R71, Y88, Y124, K130, D131, W132, F134 and K135.
(ii) Data fitting of the equilibrium binding experiments were carried out using a hyperbolic function. The authors should provide greater details in the Methods about this "hyperbolic function".
(iii) Page 3, the reported Kd values are not entirely consistent with those reported by Zhuang et al., as stated by the authors. The reported Kd for the interaction between MATH and MacroH2A is between ~8-40 times stronger (depending on which Kd is used in Table 1) than that reported in ref. 13 (63.6 micromolar). The authors should discuss this work with this previous work providing some rationale for the difference observed (or not, in the case of the puc Kd).
(iv) Page 4, fitted time resolved fluorescence data to a single exponential equation should be presented as supplementary data and define the exponential equation in the Methods with detail of how the fitting was carried out.
(v) Page 4, given the importance of the potential finding that MacroH2A and PTEN bind MATH via a non-two-state model, the authors should provide additional experiments and argument to support this suggestion, i.e., how are these peptides binding? This is a potentially important finding. Additionally, would fitting the data then differ to that used for the puc:MATH interaction, or does fitting of MacroH2A and PTEN binding to MATH require a different model than the two-state model? This should be explained in much greater detail in the results and methods.
(vi) Section 2.3, alanine mutational analysis has been reported in ref. 13 (see Fig. 3). Interestingly the previously reported Kd values for MATH mutants Y88A, Y124A, D131A and F134A are > 25 micromolar for interaction with puc, which are overall weaker affinity than those reported in this study. The authors may wish to comment on this difference. [Note in Table 2, Y125A should be Y124A].
(vii) Label residues colored red and blue in Fig. 5 with residue number information.
(viii) The discussion on the binding mechanism (page 8, above Fig. 6) is rather speculative and not supported by the experimental data. For example, in ref. 13, structural data shows subtle differences in side chain orientations of MATH domain residues involved in peptide recognition among the different peptide-domain complexes. Thus, conformational changes to the MATH domain appear to be also responsible for binding besides a proposed multi-step recognition event. Additionally, no apo-MATH structural data is presented (see Fig. 6) to indicate that no structural change occurs on peptide recognition. The data does not provide answers on whether this binding event is conformational selection or induced fit. Therefore, the authors should consider this in their wording of this part of the discussion.
Minor corrections of English grammar are required.
Round 2
Reviewer 2 Report
The authors have answered the comments raised satisfactorily.
Editing of the manuscript is recommended.